# Combining Classic and Novel Neutrophil-Related Biomarkers to Identify Non-Small-Cell Lung Cancer

**DOI:** 10.3390/cancers16030513

**Published:** 2024-01-25

**Authors:** Yunzhao Ren, Qinchuan Wang, Chenyang Xu, Qian Guo, Ruoqi Dai, Xiaohang Xu, Yuhao Zhang, Ming Wu, Xifeng Wu, Huakang Tu

**Affiliations:** 1Department of Big Data in Health Science, School of Public Health, Center of Clinical Big Data and Analytics, The Second Affiliated Hospital, Zhejiang University School of Medicine, 866 Yuhangtang Rd., Hangzhou 310058, China; 11918143@zju.edu.cn (Y.R.); wangqinchuan@zju.edu.cn (Q.W.); 22118855@zju.edu.cn (C.X.); 22118920@zju.edu.cn (Q.G.); 22118915@zju.edu.cn (R.D.); 12018344@zju.edu.cn (X.X.); zhangyuhao0580@outlook.com (Y.Z.); 2The Key Laboratory of Intelligent Preventive Medicine of Zhejiang Province, 866 Yuhangtang Rd., Hangzhou 310058, China; 3Department of Surgical Oncology, The Affiliated Sir Run Run Shaw Hospital, Zhejiang University School of Medicine, 3 East Qingchun Rd., Hangzhou 310016, China; 4Department of Thoracic Surgery, The Second Affiliated Hospital, Zhejiang University School of Medicine, 88 Jiefang Rd., Hangzhou 310009, China; iwuming22@zju.edu.cn; 5Cancer Center, Zhejiang University, 866 Yuhangtang Rd., Hangzhou 310058, China

**Keywords:** non-small-cell lung cancer, neutrophils, biomarkers, Interleukin-6, Interleukin 1 receptor antagonist, diagnosis

## Abstract

**Simple Summary:**

This study explored the predictive value of neutrophils and neutrophil-related biomarkers as auxiliary diagnosis biomarkers of NSCLC in an ongoing large cohort. IL-6 and IL-1RA were identified as independent risk factors for NSCLC. These findings can improve the predictive performance beyond epidemiological variables and classic neutrophil-related biomarkers in identifying NSCLC.

**Abstract:**

Background: Recent studies have revealed that neutrophils play a crucial role in cancer progression. This study aimed to explore the diagnostic value of neutrophil-related biomarkers for non-small-cell lung cancer (NSCLC). Methods: We initially assessed the associations between classic neutrophil-related biomarkers (neutrophil-to-lymphocyte ratio (NLR), absolute neutrophil counts (NEU), absolute lymphocyte counts (LYM)) and NSCLC in 3942 cases and 6791 controls. Then, we measured 11 novel neutrophil-related biomarkers via Luminex Assays in 132 cases and 66 controls, individually matching on sex and age (±5 years), and evaluated their associations with NSCLC risk. We also developed the predictive models by sequentially adding variables of interest and assessed model improvement. Results: Interleukin-6 (IL-6) (odds ratio (OR) = 10.687, 95% confidence interval (CI): 3.875, 29.473) and Interleukin 1 Receptor Antagonist (IL-1RA) (OR = 8.113, 95% CI: 3.182, 20.689) shows strong associations with NSCLC risk after adjusting for body mass index, smoking status, NLR, and carcinoembryonic antigen. Adding the two identified biomarkers to the predictive model significantly elevated the model performance from an area under the receiver operating characteristic curve of 0.716 to 0.851 with a net reclassification improvement of 97.73%. Conclusions: IL-6 and IL-1RA were recognized as independent risk factors for NSCLC, improving the predictive performance of the model in identifying disease.

## 1. Introduction

Worldwide, lung cancer stands as the principal cause of cancer-related deaths. In 2020, approximately 2.2 million new lung cancer cases were diagnosed, with 1.8 million fatalities. Over 85% of all lung cancer patients have non-small-cell lung cancer (NSCLC) [1].

Neutrophils, the most abundant cells in human blood circulation, are recently identified as a crucial player during carcinogenesis [2]. Neutrophils are the primary cell type during the acute inflammatory response, rapidly recruited to the affected tissue through a multi-step cascade [3], and capable of eliminating pathogens through diverse mechanisms, including phagocytosis, release of antimicrobial proteins, and formation of neutrophil extracellular traps (NETs) [4]. During the resolution of inflammation or in an anti-inflammatory state, the involvement of neutrophils is also significant. Their phagocytic activity aids in the clearance of dead cells and bacteria, thereby contributing to the elimination and reconstruction of the affected area. This can be attributed to the essential functions of several proteases expressed by neutrophils, including MMP9 and VEGFA, in tissue repair, remodeling, and angiogenesis [5,6]. The persistent infiltration of neutrophils causes chronic inflammation, which in turn leads to tissue damage and plays a significant role in the onset of cancer. This lasting and unresolved tissue inflammation is a characteristic feature of the tumor microenvironment [6]. It has been proven that neutrophils can modulate tumor progression during the onset and growth of cancer, possessing both pro-tumoral and anti-tumoral functions [5,7]. Based on the different mediators of cancer cells and the tumor microenvironment, neutrophils can be polarized into different activation states, thereby playing distinct functions in alteration of tumor progression [8]. For instance, neutrophils could promote tumorigenesis via reactive oxygen species (ROS) induced DNA damage in a lung cancer model [9], whereas it could also attack tumor cells by a neutrophil-dependent cytotoxic effect via a phagocytosis signaling of signal regulatory protein-α (SIRPα)–CD47 interaction [10,11]. However, given the multifaceted roles and varied phenotypes of neutrophils, the current research on the connection between neutrophils and lung cancer is limited.

Neutrophils and tumor-associated neutrophils (TANs) are associated with key features of resistance to immune checkpoint inhibition, such as adaptive immune cell polarization and suppression, tumor neoangiogenesis, immune exclusion, and cancer-cell-intrinsic characteristics [11,12,13,14,15]. Also, multiple studies have shown that neutrophil-to-lymphocyte ratio (NLR) can predict the clinical response of ICI treatment [12,13]. In clinical practice, the balance of inflammatory and immune responses is frequently reflected by the NLR in peripheral blood [14]. It has emerged as a prognostic factor for the survival and treatment responses in several cancers [15]. NLR is also reported as a promising predictive biomarker for immune checkpoint inhibition in NSCLC patients [14]. However, the diagnostic value of NLR in NSCLC and its underlying mechanisms are yet to be extensively studied.

Depending on the context, neutrophils play a dual role in tumor development. They promote inflammation through the release of ROS or proteases, and promote tumor dissemination and metastasis by facilitating immune suppression, angiogenesis, cancer cell motility, and epithelial-to-mesenchymal transition (EMT) [8,16]. Recent investigations have highlighted the critical role of NETs in tumor initiation and metastasis [16]. Meanwhile, neutrophils can restrict cancer growth through cytotoxic activities, such as the release of iNOS, which exerts cytotoxic effects on cancer cells. Moreover, they can inhibit tumor metastasis through mechanisms mediated by H202 or TSP1 [5,16]. TANs could impact anti-tumor immunity via secreting cytokines crosstalk with CD8^+^ T cells. TANs could produce proinflammatory factors such as monocyte chemoattractant protein-1, interleukin-8, macrophage inflammatory protein-1 alpha, interleukin-6 (IL-6), and anti-inflammatory interleukin 1 receptor antagonist (IL-1RA), thereby bolstering anti-tumor immunity in early-stage lung cancer [17]. Also, other cytokines, including interleukins (ILs), colony-stimulating factor (CSF), interferon (IFN), and chemokines, demonstrated significant associations with tumorigenesis in terms of modulating intercellular interactions and regulating immune responses [18,19,20]. Understanding the multifunctionality of neutrophils, their diverse phenotypes in different environments, and the potential for reprogramming has significant implications for understanding cancer initiation and progression [5,21]. Monitoring neutrophil-associated cytokines throughout disease progression may serve as a predictive tool for disease onset and development. However, until now, no comprehensive study has been implemented to systematically illustrate their impact on lung cancer risk.

We carried out a multi-phase study to investigate the potential diagnostic value of neutrophil-related biomarkers in NSCLC development. In the first phase, we explored the predictive efficacy of NLR as an auxiliary biomarker in diagnosing NSCLC in a large cohort encompassing lung cancer patients and healthy controls. In the second phase, we further included 132 patients from the lung cancer cohort and 66 matched healthy individuals and investigated the association between eleven novel blood neutrophil-related biomarkers and the risk of NSCLC. In the third phase, we developed predictive models incorporating classic and novel neutrophil-related biomarkers and clinical variables for the diagnosis of NSCLC.

## 2. Materials and Methods

### 2.1. Study Population and Data Collection

In the first phase of this study, a case–control design was adopted to explore the associations between classic blood neutrophil-related biomarkers (NLR, absolute neutrophil counts (NEU), absolute lymphocyte counts (LYM)), and the risk of NSCLC. From an ongoing cohort study begun in 2020 at The Second Affiliated Hospital Zhejiang University School of Medicine (SAHZU), a total of 3942 NSCLC patients were recruited. And 6791 healthy controls were drawn from the concurrently recruited healthy controls who had health check-up examinations in the general practice clinic at the SAHZU. In the second phase of the study, we further measured 11 novel blood neutrophil-related biomarkers in 132 cases and 66 healthy controls, which were individually matched on sex and age (±5 years).

The inclusion criteria for the case group are as follows: (1) with clinically and histopathologically confirmed NSCLC; (2) the patient has provided informed consent or waived consent; and (3) with data on classic blood neutrophil-related biomarkers. The exclusion criteria for the case group are as follows: (1) multiple cancers; and (2) any previous treatment undertaken by the patient at the point of enrollment.

The inclusion criteria for the healthy control group are as follows: (1) regular health examination participants from the SAHZU; and (2) participants have provided informed consent or waived consent. The exclusion criteria are as follows: (1) suffering from severe lung disease; and (2) diagnosis of any malignant neoplasm.

This study has received approval from the Institutional Review Board of SAHZU. Clinical pathological information was derived from detailed chart reviews. Concentrations of NEU, LYM, and carcinoembryonic antigen (CEA) in the blood were quantified at NSCLC diagnosis for cases or during routine health examinations for the controls. The staging of lung cancer patients was carried out by the attending physicians and pathologists in accordance with the NCCN Clinical Practice Guidelines Non-small-cell lung cancer v1, 2022.

Staff members gathered epidemiological information through face-to-face interviews. The participants’ weight, height, history of hypertension (yes or no), history of diabetes (yes or no), and smoking status were recorded upon enrollment. The body mass index (BMI) was derived by taking the ratio of weight to the square of the height (kg/m^2^). According to the WHO guidelines, we divided BMI into two categories: underweight/normal (<25 kg/m^2^) and overweight/obese (≥25 kg/m^2^). The categorization of smoking status depends on whether the subject had ever smoked (defined as having smoked at least 100 cigarettes in their lifetime) [22,23].

### 2.2. Detection of Novel Neutrophil-Related Biomarkers via Luminex Assays

In stage 2, venous blood samples of 20 mL were collected from 198 participants using ethylenediaminetetraacetic acid tubes and promptly delivered to the SAHZU laboratory. Prior to the initiation of the experiment, plasma was isolated, divided into aliquots, and preserved at −80 °C. Plasma samples were defrosted on ice [23]. The concentrations of 11 novel neutrophil-related biomarkers (IL-1α, IL-1β, IL-1RA, IL-6, IL-17, G-SCF, GM-CSF, CXCL2, CXCL5, IFN-α, and S100B) in the plasma samples were quantitatively determined utilizing the Luminex Discovery Assay—Human Premixed Multi-Analyte Kit (R&D Systems, Minneapolis, MN, USA, LXSAHM-11), strictly adhering to the protocol provided by the manufacturer [24]. To ensure the reliability and accuracy of the measurements, each sample was assayed in duplicates on a 96-well plate using a Luminex FLEXMAP 3D system (Luminex Corp, Austin, TX, USA), utilizing undiluted plasma. Each plate incorporated both positive and negative controls, as well as samples for the purpose of generating the standard curve [23,25]. Laboratory personnel were completely blinded to the case and control status. The assay was performed in alignment with the manufacturer’s instructions.

We selected 11 novel blood neutrophil-related biomarkers based on comprehensive literature reviews and the feasibility of using Luminex assays. G-CSF is a critical regulatory agent in the biological genesis of neutrophils, with its receptors being expressed throughout the entire bone marrow lineage, ranging from early stem cells and progenitor cells to the mature status of neutrophils [26]. GM-CSF and IL-6 are both acknowledged as cytokines involved in granulocyte formation and neutrophil proliferation in various types of cancer [26]. IL-1, once activated, functions as a robust pro-inflammatory cytokine locally, instigating vasodilation and recruiting monocytes and neutrophils to the stress location [27]. The generation of active IL-1β is facilitated by inflammasomes or neutrophil proteases through cleaving pro-IL-1β, a process mediated by caspase-1 [27]. IL-1α triggers sterile inflammation through the induction of neutrophil mobilization in reaction to cell death. IL-1 RA, which can be released by neutrophils, binds and blocks IL-1 Receptor Type 1, competitively inhibiting the pro-inflammatory action of IL-1 [28]. IL-17, generated by neutrophils, T cells, innate lymphoid cells, natural killer cells, macrophages, and so on, exerts a crucial role in the recruitment of neutrophils [29]. CXCL2 plays a crucial role in neutrophil recruitment by interacting with CXCR2 on neutrophils [30]. CXCL5 can enhance the immunosuppressive features of the tumor microenvironment by stimulating immune cell migration to the tumor and recruiting vascular endothelial cells for angiogenesis, thus promoting tumor progression [31]. By targeting both tumor cells as well as immune cells, type I interferons have demonstrated a pivotal role in inhibiting tumor growth [32,33]. The S100 family proteins also hold vital value in natural immunity and act as mediators in inflammatory responses. Neutrophils, among other immune cells, can produce considerable amounts of S100 A8/A9, which control inflammation by triggering the discharge of cytokines and ROS. S100B is one of the most active members of the S100 family [34,35]. Among the 11 biomarkers, 10 (except S100B) are produced by neutrophils under certain circumstances. In general, IL-1RA, IFN-α, G-CSF, and GM-CSF are considered anti-tumor biomarkers, while the remaining factors are considered pro-tumor biomarkers.

### 2.3. Statistical Analysis

Categorical variables are characterized by frequencies with percentages. Statistical differences between groups were compared using chi-square tests or Fisher’s exact probability method. Continuous variables are depicted using mean ± standard deviation (SD) or median [25th and 75th percentiles (Q1–Q3)] depending on the distribution type. A comparison of groups for statistical differences was conducted using *t*-tests (for normal distributions), Kruskal–Wallis tests or Wilcoxon rank-sum tests (for non-normal distributions). For the matched paired samples in stage 2, the comparison of measurement data between the two groups was conducted using the paired samples *t*-test or non-parametric test, while the comparison of categorical data was performed using the paired chi-square test or non-parametric test. Furthermore, we conducted a sensitivity analysis in stage 1 with cases and controls being individually matched on sex and age (±5 years).

All biomarkers were further processed as categorical variables to minimize skewness. Using the median value in the control group as the cutoff, the values of NLR, NEU, LYM, IL-6, CXCL2, IL-1RA, IL-1α, and CXCL5 were classified into low and high groups. Meanwhile, the values of S100B and GM-CSF were divided into low and high groups, with the experimental detection limit serving as the cutoff because a substantial of individuals had levels under the experimental detection limit. The CEA value was divided into normal and abnormal groups using the threshold of 5 ng/mL.

In stage 2, given the matched case–control design, the associations between novel neutrophil-related biomarkers and NSCLC risk were examined by conditional logistic regression. We first performed a univariate analysis for the 11 measured novel markers, followed by a multivariate analysis adjusting for epidemiological variables, NLR, and CEA, founded on the univariate analysis results and prior knowledge. The model outputs the odds ratio (OR) and its 95% confidence interval (CI) to estimate the strength of the association between the novel biomarkers and NSCLC risk.

In stage 3, we developed the predictive models by sequentially adding variables of interest in the study population from stage 2. The risk prediction model was initially built based on health history (model 1: BMI + smoking status). Then, we included relevant clinical biomarkers (model 2: model 1 + NLR + CEA). Further, two newly identified novel blood neutrophil-related biomarkers were incorporated (model 3: model 2 + IL-6 + IL-1RA). We applied the receiver operating characteristic (ROC) curve and the area under the curve (AUC) to evaluate the discriminative capacity of different models for NSCLC risk. Delong’s test was used to test whether the differences in model performance across different models. Additionally, the true positive rate (TPR) and false positive rate (FPR) were calculated. To evaluate whether the predictive performance of models was enhanced after the addition of more predictors, we calculated the net reclassification improvement (NRI) and integrated discrimination improvement (IDI) metrics.

The collection of data and the presentation of tables were conducted in Excel (Microsoft Office 2021 version). R software (v4.2.1) was utilized for all data analysis and visualization. Specifically, the R software packages “readxl”, “readr”, “plyr”, “dplyr”, “data.table”, “tableone”, “table1”, “stringr”, “forcats”, “reshape2”, “broom”, “tidyverse”, and “tidyr” were used for data preparation (including reading, cleaning, data transformation, etc.). The R package “Hmisc” was applied for the correlation analysis and the “pROC” for the ROC curve. The restricted cubic spline analysis used the R packages “rms”, “Hmisc”, “car”, and “smoothHR”. The comparative analysis was carried out by the R package “rstatix”. The values of NRI and IDI were calculated using the R packages “nricens” and “PredictABEL”. In order to visualize our results, the R packages “pheatmap”, “ggpubr”, “ggplot2”, “ggthemes”, “grid”, “gridExtra”, and “forestploter” were employed. The R packages used for the logistic regression analysis were “pubh”, “rms”, “survival”, “car”. All R packages and their instructions used in our study can be found in the link (https://cran.r-project.org/web/packages/available_packages_by_name.html (accessed on 25 September 2023). All statistical tests were two sided, setting the significance level at 0.05.

## 3. Results

### 3.1. Stage 1

Appendix A presents the baseline characteristics of all subjects in stage 1. More than half of the healthy participants were female, while nearly two-thirds of patients were male. The age of the subjects increased with disease status and severity. The proportion of smokers was higher among patients with invasive adenocarcinoma (IAC), which includes early-stage and late-stage NSCLC. The majority of patients did not present with either lymph node metastasis or distant metastasis at the time of enrollment.

The restricted cubic spline analysis revealed a positive association between NEU and NSCLC risk, while an inverse association was seen with LYM (Figure 1B,C). NLR showed a strong non-linear positive association with the risk of NSCLC (Figure 1A).

The unconditional univariate logistic regression analysis indicated significant correlations between classic blood biomarkers and NSCLC risk (Appendix A and Figure 2). Elevated levels of NLR and NEU were associated with increased risk of NSCLC (OR_NLR_ = 2.983 (95% CI: 2.737, 3.252), OR_NEU_ = 1.755 (95% CI: 1.620, 1.902)), whereas a high blood concentration of LYM was associated with decreased risk of NSCLC (OR = 0.434, 95% CI: 0.399, 0.471). After adjusting for age, sex, BMI, and smoking status, compared with the low-level groups, the high-level groups of NLR and NEU were associated with higher risks of NSCLC (OR_NLR_ = 2.608 (95% CI: 2.333, 2.918), OR_NEU_ = 2.222 (95% CI: 1.994, 2.477)), while the higher LYM was associated with a lower NSCLC risk (OR_LYM_ = 0.650 (95% CI: 0.583, 0.724)) (all *p* values were less than 0.05). Stratified analysis by age, sex, BMI, and smoking status also revealed significant and highly consistent associations between these three indicators and NSCLC risks (Figure 2). In addition, our sensitivity analysis indicated the results were notably consistent after matching the cases and controls on age and sex.

We further analyzed the differences in NEU, LYM, and NLR across the control, carcinoma in situ (Tis), early-stage, and late-stage groups (Table 1). The concentrations of NEU in cancer groups were significantly higher while LYM was lower in comparison to the control group. The NLR was observed to be higher in the NSCLC groups. However, the difference between the Tis and early-stage groups among these three indicators was not significant. Compared to early-stage patients, late-stage patients had higher NLR and NEU levels, while the difference was not significant in LYM level. Figure 1D–F graphically illustrates the differences across those groups.

Appendix A depicts the comparison of classic blood biomarkers across various clinical features among the healthy controls.

### 3.2. Stage 2

Appendix A shows the host characteristics of the subset of the study participants with measurements of 11 novel blood neutrophil-related biomarkers. This phase included a total of 198 participants, 132 of whom were NSCLC patients and 66 were healthy controls. In the NSCLC group, there were 90 cases of invasive cancer (68.18%). Subsequent analyses did not include IL-1β, IFN-α, IL-17, and G-CSF due to excessive missing assay values.

The differences in NLR, NEU, and LYM between cases and controls in stage 2 were consistent with those in stage 1 (Appendix A). In terms of novel blood neutrophil-related biomarkers, the IL-6 and IL-1RA levels in the controls were significantly lower than in the cases (*p* < 0.001) (Appendix A and Appendix A).

Appendix A presents the pairwise correlations among the novel blood neutrophil-related biomarkers. IL-1α and CXCL5 have a correlation coefficient of 0.9, indicating a strong correlation. The correlation coefficients for the remaining factors were all below 0.5, indicating weaker correlations.

In univariate analysis (Table 2), among the analyzed novel blood neutrophil-related biomarkers, higher levels of IL-6 and IL-1RA were significantly associated with increased risk of NSCLC, with OR values of 9.339 (95% CI: 3.882, 22.646) and 7.535 (95% CI: 3.293, 17.244), respectively. In multivariate conditional logistic regression analysis conducted on IL-6, CXCL2, IL-1RA, IL-1α, CXCL5, S100B, and GM-CSF, we found that after adjusting for BMI, smoking status, NLR, and CEA, higher plasma levels of IL-6 and IL-1RA were associated with substantially elevated risk of NSCLC. The risk of NSCLC in individuals with higher plasma levels of IL-6 was 10.687 times (95% CI: 3.875, 29.473) that of the low-level group, and similarly, the risk was 8.113 times (95% CI: 3.182, 20.689) higher for those with high levels of IL-1RA. The forest plots of the univariate and multiple conditional analysis are shown in Appendix A.

### 3.3. Stage 3

Appendix A shows the comparison of the discrimination ability of different predictive models. In Model 1, which only included the epidemiological predictors (BMI and smoking status), the AUC value was 0.603 (95% CI: 0.527, 0.678). After adding NLR and CEA, the AUC value of Model 2 increased to 0.716 (95% CI: 0.637, 0.794). Upon the addition of IL-6 and IL-1RA to Model 2 to create Model 3, the best performance was achieved with an AUC value of 0.851 (95% CI: 0.793, 0.908), a TPR of 0.856, and a FPR of 0.333. The differences in AUCs from the three models were statistically significant. The visualized ROC curves for Models 1, 2, and 3 are shown in Figure 3.

We employed NRI and IDI to measure the improvement of the predictive performance of models after the inclusion of new risk factors, enabling the comparison among models (Table 3). Initially, we compared Model 2 (Model 1 + NLR + CEA) with Model 1 (baseline model incorporating only epidemiological indicators: BMI + smoking status). The results indicated a noteworthy enhancement in the predictive performance of Model 2 in contrast to Model 1. The NRI reached 59.85% (95% CI: 0.331, 0.935), and the IDI was 0.128 (95% CI: 0.079, 0.176), both differences being statistically significant. By incorporating IL-6 and IL-1RA into Model 2, the NRI for Model 3 increased to 97.73%, signifying that the integration of IL-6 and IL-1RA could escalate the accurate reclassification proportion of the model by 97.73% (95% CI: 0.667, 1.279). The extent of improvement was also significantly increased, with an IDI of 0.198 (95% CI: 0.137, 0.259).

## 4. Discussion

Our findings indicated a significant association between NLR and NSCLC risk. Plasma IL-6 and IL-1RA also emerged as independent risk factors for NSCLC. These mean high levels of NLR, IL-6, and IL-1RA signaling a dramatically increased NSCLC risk. Moreover, we established a model incorporating NLR and novel blood neutrophil-related biomarkers to aid in predicting NSCLC diagnosis in the clinic, outperforming models reliant on the classic cancer marker CEA. Thus, our research underscores the importance of neutrophil-related biomarkers in predicting NSCLC risk, offering valuable assistance in clinical diagnosis.

NLR, a parameter derived from the NEU divided by the LYM, is one of the most widely investigated features based on blood cell counts and serves as an indicator of systemic inflammation [36]. NLR captures the balance between the detrimental effects of increased neutrophils and the beneficial roles of adaptive immunity mediated by lymphocytes [37]. In the field of cancer research, NLR serves as an effective indicator of the dynamic balance between pro-tumor and anti-tumor responses in the body. Most published studies have mainly focused on the correlation between NLR and the prognosis of NSCLC [36,37,38,39,40], with little attention paid to its association with disease risk. Therese Haugdahl Nøst et al. conducted a study on approximately 440,000 participants based on data from the UK Biobank, evaluating the longitudinal relationships between four systemic inflammation indicators (NLR, systemic immune-inflammation index, platelet-to-lymphocyte ratio, lymphocyte-to-monocyte ratio) and the risk of 17 cancer sites diagnosed clinically in the years preceding the study [41]. The study found that NSCLC patients exhibited higher NLR values, which was particularly apparent in the year before diagnosis, possibly driven by an elevated neutrophil count.

IL-6, a multifunctional cytokine, exhibits both pro-inflammatory and anti-inflammatory properties [42]. Under cancerous conditions, IL-6 can participate in various processes such as tumor formation, cancer cell proliferation, epithelial–mesenchymal transition, interactions between tumor cells and the matrix environment, tumor dissemination, and drug resistance. By releasing chemokine receptors including CXCR3/4 and CCR5/7, and secreting pro-inflammatory cytokines, notably IL-6, TANs play a pivotal role in the pathogenesis of lung cancer [43]. Currently, the predictive capabilities evaluation of IL-6 in cancer mostly focuses on treatment outcomes or survival status [44,45,46,47,48,49,50], making it one of the most discussed prognostic markers for NSCLC patients. However, studies dedicated to evaluating associated risks remain sparse. In a case–control study conducted at the National Cancer Institute in Maryland, Sharon R. Pine and colleagues analyzed the association between IL-6 and lung cancer in six pairs of patients and controls. They discovered a notable association between lung cancer and the highest quartile of serum IL-6 levels (OR = 3.29, 95% CI: 1.88–5.77) [51]. In a subsequent verification within the prospective Prostate, Lung, Colorectal, and Ovarian Cancer Screening Trial, they found that an increase in IL-6 levels was exclusively connected to lung cancer cases diagnosed within two years of blood sampling. A nested case–control study including 224 cases and 644 controls indicated higher blood IL-6 was associated with rising hepatocellular carcinoma (HCC) risk [52]. The relative risk was 5.12 (95% CI: 1.54–20.1) for HCC in the top tertile of IL-6 levels. This result was not influenced by variables such as hepatitis virus infection, lifestyle-related elements, and radiation exposure. Scholars have underscored the importance of early monitoring of IL-6 levels. Further gene expression analysis showed an increased expression of IL-6 in patients compared to controls. The latest research on the Lung Cancer Cohort Consortium, an international cohort with over two million participants from North America, Europe, Asia, and Australia, identified 36 proteins independently and reproducibly associated with the imminent risk of being diagnosed with lung cancer. IL-6 was found a robust correlation with lung cancer within the first year following diagnosis (OR = 2.56, 95% CI: 1.92–3.41) [34]. Therefore, the measurement of IL-6 in plasma can serve as an early auxiliary diagnostic indicator for NSCLC, providing predictive value for clinical work.

IL-1RA, a component of the IL-1 family, operates as a competitive binding factor that can inhibit the signal cascade response and suppress the pro-inflammatory signal transduction activated by IL-1α and IL-1β [53]. IL-1RA is commonly generated by the cells that concurrently produce IL-1α or IL-1β, notably monocytes, macrophages, dendritic cells, neutrophils, and so on. An increase in IL-1 production is often paired with a raised IL-1RA level. Even though IL-1RA does not trigger a biological response, it is described as an anti-inflammatory molecule due to its capacity to impede the pro-inflammatory activities of IL-1α and IL-1β [53]. The expression of IL-1RA has been investigated in many human diseases, such as inflammatory diseases, immune-related diseases, and numerous kinds of cancer. Generally, IL-1RA in cancer is considered to exert a tumor-suppressive effect due to its ability to inhibit pro-tumor cytokines [54]. Post myocardial infarction, animals with high expression of IL-1RA displayed decreased symptoms of inflammation, less neutrophil infiltration, and reduced ventricular expansion [55]. In a range of diseases, increased levels of circulating IL-1RA have been reported, such as chronic arthritis, inflammatory bowel disease, rheumatoid arthritis, and Acute Respiratory Distress Syndrome [56]. Synthesized research suggested that the balance between IL-1RA and the pro-inflammatory cytokine IL-1 was associated with increased risks of various cancers, including NSCLC [57]. Elevated serum IL-1RA concentrations had been observed in patients with Hodgkin’s disease, lung cancer, colorectal cancer, cervical cancer, and endometrial cancer, underscoring the critical function of IL-1RA throughout the formation and advancement of tumors [54]. In 2013, a nested case–control study was performed on 526 lung cancer patients and 592 control subjects [20]. Using the Luminex beads-based experimental method, more than 70 serum inflammation markers were examined. The researchers found a correlation between increased serum IL-1RA levels and a decreased risk of lung cancer (OR = 0.71, 95% CI: 0.51–1.00). Genetic research in humans also showed that the genetic variability of IL-1α, IL-1β, and IL-1RA correlated with elevated risks of tumors, including NSCLC [58]. Although IL-1RA was discovered almost concurrently with IL-1α and IL-1β, its value in cancer remains relatively uncertain compared to the other two. Current research on IL-1RA in tumors mostly reports its anti-inflammatory role. However, some scholars have reported that the role of IL-1RA in cancer is not limited to suppressing inflammation, but it can also promote the growth of malignant tumors [53].

We acknowledge that there are several limitations. First, several novel factors that were to be tested in the samples inherently had low expression levels, which made their detection difficult. They were excluded from data analysis due to numerous undetected values, despite our meticulous handling of the samples, including storage at −80 degrees Celsius, prevention of repeated freeze–thaw cycles, and strict adherence to the instructions during the procedures. Second, as a cross-sectional study, we were unable to determine causality. Further rigorous validation work along with an increase in sample size is needed. Third, our research focused on studying the NSCLC population. However, the significant variations among different cancers and subtypes limit the generalizability of our findings.

## 5. Conclusions

Our study suggests that IL-6 and IL-1RA play key roles in lung carcinogenesis and progression. NLR, IL-6, and IL-1RA in the blood can serve as biomarkers for diagnosis of NSCLC. Combining these with patients’ clinical features and tumor markers (for example, CEA) may enhance the effectiveness of diagnosing NSCLC, potentially providing heightened early warning at pre-diagnosis and diagnosis.

## Figures and Tables

**Figure 1 cancers-16-00513-f001:**
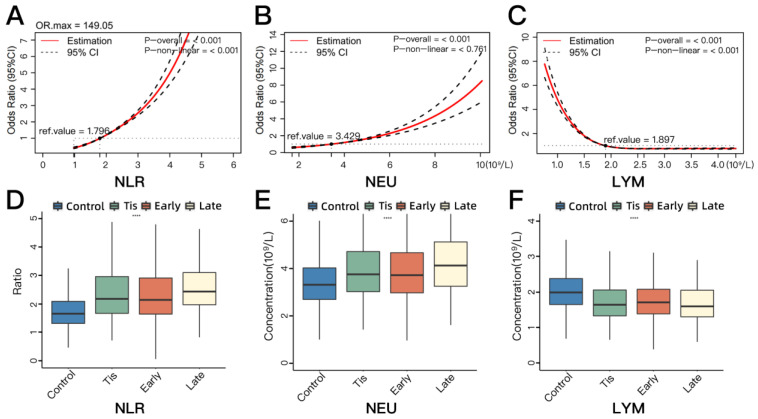
The associations between classic neutrophil-related blood biomarkers (NLR, NEU, and LYM) and NSCLC risk based on restricted cubic splines (**A**–**C**) and the inter-group comparisons of classic biomarkers’ distribution across four groups (control, Tis, early-stage NSCLC, and late-stage NSCLC) (**D**–**F**). (**A**) The associations between the plasma level of NLR and NSCLC risk. (**B**) The associations between the plasma concentration of NEU and NSCLC risk. (**C**) The associations between the plasma concentration of LYM and NSCLC risk. (**D**) The distribution of plasma concentration of NLR across four groups. (**E**) The distribution of plasma concentration of NEU across four groups. (**F**) The distribution of plasma concentration of LYM across four groups. Tis, carcinoma in situ; NSCLC, non-small-cell lung cancer; NLR, neutrophil-to-lymphocyte ratio; NEU, absolute neutrophil counts; LYM, absolute lymphocyte counts; OR, odds ratio; CI, confidence interval. Early stage included stage 1 and 2 diseases. Late stage included stage 3 and 4 diseases. The reference value (ref. value) means the level of the biomarker when the corresponding OR is 1 (the horizontal dotted line). The *p*-overall indicates the statistical significance of the association between the biomarker and NSCLC risk, with *p*-overall < 0.05 indicating a statistically significant association. The *p*-non-linear value indicates whether there is a nonlinear relationship between the biomarker and NSCLC risk, with *p*-non-linear < 0.05 indicating that the association between the biomarker and NSCLC risk could not be fully explained by a linear relationship; in other words, there was a non-linear association. **** *p* < 0.0001.

**Figure 2 cancers-16-00513-f002:**
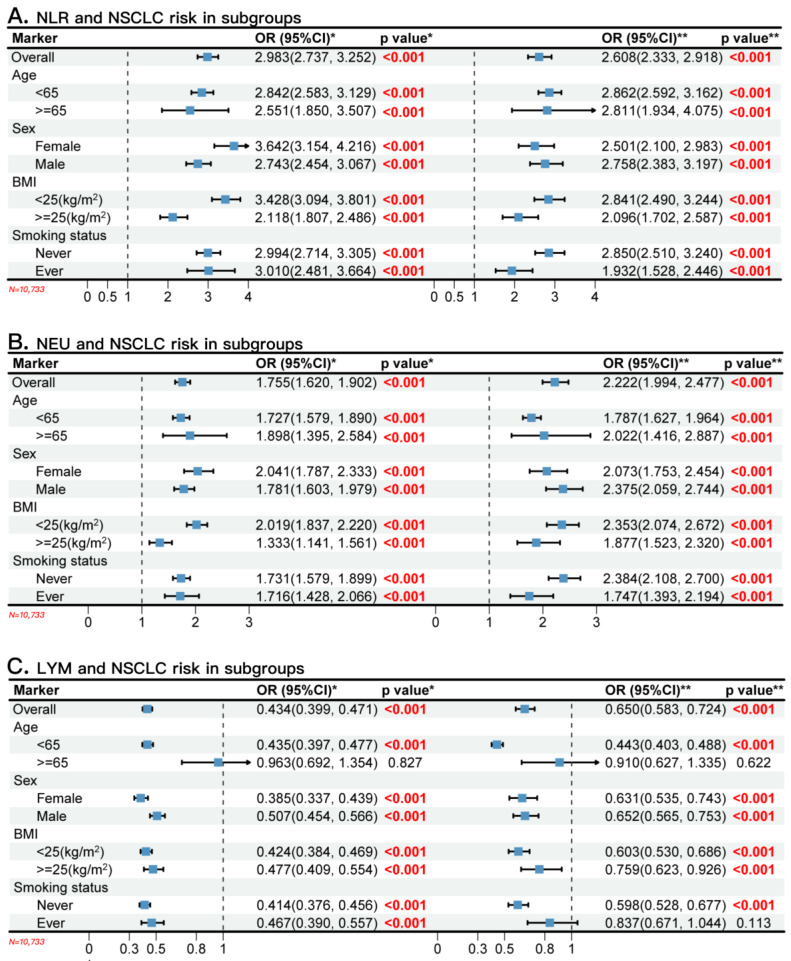
The overall and stratified associations between classic blood neutrophil-related biomarkers (NLR, NEU, and LYM) and NSCLC risk in a large clinical cohort. (**A**) The univariate (**left**) and multiple (**right**) logistic regression analysis of NLR and the stratified analysis. (**B**) The univariate (**left**) and multiple (**right**) logistic regression analysis of NEU and the stratified analysis. (**C**) The univariate (**left**) and multiple (**right**) logistic regression analysis of LYM and the stratified analysis. NLR, neutrophil-to-lymphocyte ratio; NEU, absolute neutrophil counts; LYM, absolute lymphocyte counts; BMI, body mass index; OR, odds ratio; CI, confidence interval. * shows the OR (95%CI) and *p* values of the univariate conditional logistic regression analysis. ** shows the OR (95%CI) and *p* values of the multiple conditional logistic regression analysis.

**Figure 3 cancers-16-00513-f003:**
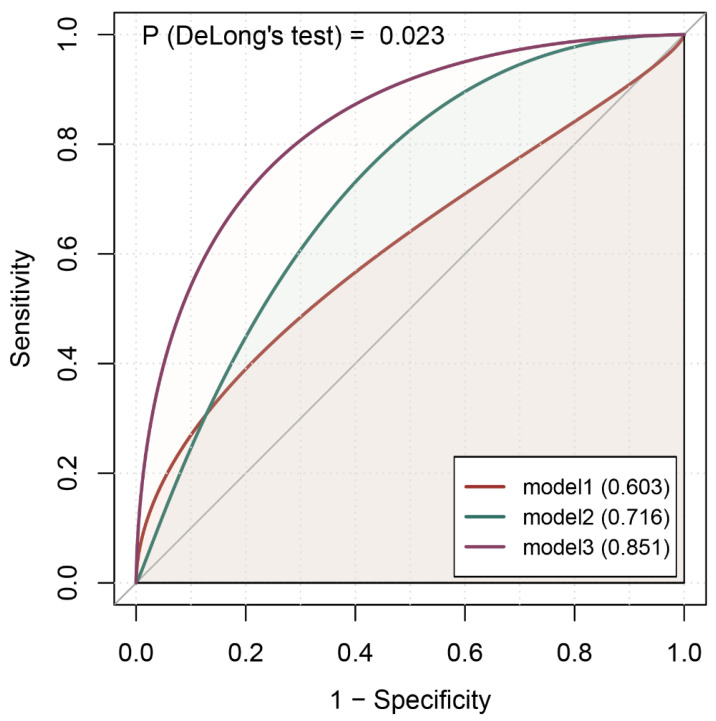
Comparison of ROC curves for different models. The AUC value of Model 1 was only 0.603 based on epidemiological variables (BMI and smoking status) data. In Model 2 (added NLR and CEA), the AUC value increased to 0.716. Model 3 (added IL-6 and IL-1RA) showed the best performance, with an AUC value of 0.851. IL-6, Interleukin 6; IL-1RA, Interleukin-1 receptor antagonist; NLR, neutrophil-to-lymphocyte ratio; CEA, carcinoembryonic antigen; BMI, body mass index; ROC, receiver operating characteristic; AUC, area under the curve.

**Table 1 cancers-16-00513-t001:** The distribution of classic blood neutrophil-related biomarkers by disease status in stage 1.

Markers	Control (*n* = 6791) Median [Q1–Q3]	NSCLC (*n* = 3942) Median [Q1–Q3]			* *p*	** *p*	*** *p*	**** *p*
		Tis (*n* = 450)	Early (*n* = 3376)	Late (*n* = 116)				
NLR	1.657 [1.315–2.090]	2.179 [1.670–2.962]	2.142 [1.644–2.911]	2.436 [1.972–3.102]	<0.001	<0.001	0.332	<0.001
NEU (10^9^/L)	3.310 [2.690–4.020]	3.750 [3.020–4.710]	3.715 [2.970–4.660]	4.120 [3.248–5.115]	<0.001	<0.001	0.670	0.005
LYM (10^9^/L)	1.990 [1.650–2.380]	1.645 [1.333–2.060]	1.710 [1.388–2.080]	1.600 [1.305–2.053]	<0.001	<0.001	0.360	0.142

Early stage indicates stage 1 and 2 diseases, late stage indicates stage 3 and 4 diseases, the staging criteria according to NCCN Clinical Practice Guidelines Non-small-cell lung cancer v1, 2022. NSCLC, non-small-cell lung cancer; Tis, carcinoma in situ; NLR, neutrophil-to-lymphocyte ratio; NEU, absolute neutrophil counts; LYM, absolute lymphocyte counts. * *p* indicates control vs. NSCLC. ** *p* indicates control vs. Tis. *** *p* indicates Tis vs. early stage. **** *p* indicates early stage vs. late stage.

**Table 2 cancers-16-00513-t002:** The associations between novel blood neutrophil-related biomarkers and NSCLC in stage 2.

Markers	Control (*n* = 66)	NSCLC (*n* = 132)	OR (95% CI) *	*p* Value *	OR (95% CI) **^, a^	*p* Value **^, a^
IL-6						
Low	34 (51.52)	15 (11.36)	1 (ref)	<0.001	1 (ref)	<0.001
High	32 (48.48)	117 (88.64)	9.339 (3.882, 22.464)		10.687 (3.875, 29.473)	
CXCL2						
Low	33 (50.00)	52 (39.39)	1 (ref)	0.151	1 (ref)	0.196
High	33 (50.00)	80 (6.61)	1.570 (0.848, 2.907)		1.824 (0.903, 3.683)	
IL-1RA						
Low	33 (50.00)	17 (12.88)	1 (ref)	<0.001	1 (ref)	<0.001
High	33 (50.00)	115 (87.12)	7.535 (3.293, 17.244)		8.113 (3.182, 20.689)	
IL-1α						
Low	33 (50.00)	64 (48.48)	1 (ref)	0.849	1 (ref)	0.615
High	33 (50.00)	68 (51.52)	1.056 (0.603, 1.850)		1.314 (0.700, 2.466)	
CXCL5						
Low	33 (50.00)	62 (46.97)	1 (ref)	0.703	1 (ref)	0.594
High	33 (50.00)	70 (53.3)	1.116 (0.636, 1.958)		1.371 (0.719, 2.613)	
S100B						
Low	42 (63.64)	70 (53.3)	1 (ref)	0.184	1 (ref)	0.531
High	24 (36.36)	62 (46.97)	1.481 (0.830, 2.645)		1.287 (0.663, 2.501)	
GM-CSF						
Low	52 (78.79)	100 (75.76)	1 (ref)	0.633	1 (ref)	0.611
High	14 (21.21)	32 (24.24)	1.191 (0.582, 2.436)		1.294 (0.563, 2.977)	

The values of IL-6, CXCL2, IL-1RA, IL-1α, and CXCL5 were divided into low and high groups based on the median value in the control group as the cutoff. The values of S100B and GM-CSF were divided into low and high groups, with the experimental detection limit serving as the cutoff. ^a^ Adjusted factors: BMI + smoking status + NLR + CEA. * shows the OR (95% CI) and *p* values of the univariate logistic regression analysis. ** shows the OR (95% CI) and *p* values of the multiple logistic regression analysis. NSCLC, non-small-cell lung cancer; BMI, body mass index; IL-6, Interleukin 6; IL-1α, Interleukin 1 alpha; IL-1RA, Interleukin-1 receptor antagonist; GM-CSF, Granulocyte-macrophage colony-stimulating factor; CXCL2, C-X-C Motif Chemokine Ligand 2; CXCL5, C-X-C Motif Chemokine Ligand 5; S100B, S100 Calcium-binding Protein B; NLR, neutrophil-to-lymphocyte ratio; CEA, carcinoembryonic antigen; OR, odds ratio; CI, confidence interval. Values are presented as *n* (%) unless otherwise specified.

**Table 3 cancers-16-00513-t003:** The comparative efficacy of predictive models.

Models	NRI	95% CI ^a^	* *p* Value	IDI	95% CI ^b^	** *p* Value
Model 1 vs. Model 2	59.85%	0.331, 0.935	<0.001	0.128	0.079, 0.176	<0.001
Model 2 vs. Model 2 + IL-6	75.00%	0.319, 1.022	<0.001	0.148	0.092, 0.203	<0.001
Model 2 vs. Model 2 + IL-1RA	61.36%	0.122, 0.962	0.003	0.122	0.068, 0.176	<0.001
Model 2 + IL-6 vs. Model 3	16.67%	−0.055, 0.779	0.423	0.050	0.012, 0.088	0.010
Model 2 + IL-1RA vs. Model 3	29.55%	−0.001, 0.866	0.227	0.076	0.033, 0.119	0.001
Model 2 vs. Model 3	97.73%	0.667, 1.279	<0.001	0.198	0.137, 0.259	<0.001

Model 1: epidemiology variables: BMI + smoking status. Model 2: epidemiology variables + NLR + CEA. Model 3: epidemiology variables + NLR + CEA + IL-6 + IL-1RA. BMI, body mass index; IL-6, Interleukin 6; IL-1RA, Interleukin-1 receptor antagonist; NLR, neutrophil to lymphocyte ratio; CEA, carcinoembryonic antigen; NRI, net reclassification improvement; IDI, integrated discrimination improvement; CI, confidence interval. ^a^ shows the 95% CI on NRI. ^b^ shows the 95% CI on IDI. * indicates the *p* value on the difference of NRI. ** indicates the *p* value on the difference of IDI.

## Data Availability

The data that support the findings of this study are available from the corresponding author upon reasonable request.

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
