# Peer review of "Combining Classic and Novel Neutrophil-Related Biomarkers to Identify Non-Small-Cell Lung Cancer"

_cancers, 2024, doi:10.3390/cancers16030513_

Round 1

Reviewer 1 Report

Comments and Suggestions for Authors

Dear editor in Chief

The authors of the manuscript entitled "Combining classic and novel neutrophil-related biomarkers to identify non-small cell lung cancer" demonstrated the possible use of novel inflammatory markers in NSCLC. 

The study design is interesting and conducted very well. The results showed a valuable use of IL-6 and IL-1RA as independent factors for assessing the NSCLC risk and prediction.

However, my concern is why the authors selected  132 cases and 66 controls from the large cohort.

Author Response

Responses to the Reviewer 1

Comments:

The authors of the manuscript entitled "Combining classic and novel neutrophil-related biomarkers to identify non-small cell lung cancer" demonstrated the possible use of novel inflammatory markers in NSCLC. The study design is interesting and conducted very well. The results showed a valuable use of IL-6 and IL-1RA as independent factors for assessing the NSCLC risk and prediction.

However, my concern is why the authors selected 132 cases and 66 controls from the large cohort.

Response: We thank the reviewer for the comment. In stage one of our study, we assessed the associations between classic neutrophil-related biomarkers (including neutrophil to lymphocyte ratio, absolute neutrophil counts, absolute lymphocyte counts) and NSCLC in all the NSCLC cases and controls available in our project, given those three classic markers are routinely tested in clinical practice. Then, we additionally measured 11 novel neutrophil-related biomarkers via Luminex Assays in 132 cases and 66 controls from the large cohort, individually matching on sex and age. We selected 132 cases and 66 controls for two main reasons. First, we selected a sample from the large cohort to be more cost-effective. Because these 11 novel biomarkers are not routinely tested in clinical practice, those biomarkers have to be additionally measured via Luminex Assays which are high-cost assays. Second, sample size calculation indicated that this sample size is reasonably sufficient. With a sample size of 132 cases and 66 controls and type I error at 0.05, our study had more than 80% power to detect an OR of 2.5, which we consider to be clinically significant.

Reviewer 2 Report

Comments and Suggestions for Authors

Major comments in this manuscript:

1. For a biological study, an important concerning is others could repeat their result. However, in this manuscript, I didn’t see the open data URL link in “Data availability”.

2. The format of Table 1 looks not tidy, please re-order it

3. Figure 3 showed the “Delong’s test”, but not mentioned in the maintext, please add it into the sentence.

4. The authors said they utilized R software (v4.2.1) and Excel (Microsoft Office 365) for all data analysis. But no detailed information (package, parameter, index, etc) was described.

Reviewer 3 Report

Comments and Suggestions for Authors

Authors investigated the efficacy of NLR as an auxiliary biomarker in diagnosing NSCLC in a large cohort of lung cancer patients and healthy controls. They then investigated the relationship between eleven novel blood neutrophil-related biomarkers and risk of NSCLC and identified Interleukin-6 (IL-6) and In-31terleukin 1 Receptor Antagonist (IL-1RA), which show a strong association with NSCLC risk. They further developed a predictive model using these two biomarkers. The findings is very interesting, and manuscript is well written except the following concerning points:

·         Readers would benefit with an explanation of immune checkpoint inhibition and its correlation with NLR and which aspects of neutrophil functions, either pro- or antitumor or immune response, to have a better understanding how NLR might be helpful for diagnosis of NSCLC.

·         The known/relevant roles of neutrophils in anti-/protumor and in anti-/pro inflammation and their crosstalk at which contents/circumstances need to be clarified/explained.

·         The relationship between neutrophil-associated cytokines and neutrophil-related biomarkers needs to be clarified. Do authors mean neutrophil-derived/produced biomarker?

·         Did author examine anti-tumor biomarkers and their relevance of NSCLC risk?

·         Figure1:

o   Figure legends need to be self-explanatory.

o   Figure scales and units are too small to read.

o   What are the ref. values, P-overall, P-non-linear? and their indications?

·         Figure 2:

o   An explanation of the scale of Risk factors and its indications is needed. So does the protective factors.

·         Table 1 is a difficult, confusing table. A better organization/presentation is helpful.

Comments on the Quality of English Language

The manuscript is well written.  

Round 2

Reviewer 2 Report

Comments and Suggestions for Authors

Accept